

# Homeostatic responses and growth of *Leymus chinensis* under incrementally increasing saline-alkali stress

Shujie Li[1,2,*], Yujin Huang[1,3,*] and Yuefen Li[1,2]

[1] College of Earth Sciences, Jilin University, Changchun, China
[2] Key laboratory of Mineral Resources Evaluation in Northeast Asia, Ministry of Land and Resources, Changchun, China
[3] Institute of Geographic Sciences and Natural Resources Research, CAS, Beijing, China
[*] These authors contributed equally to this work.

## ABSTRACT

Despite considerable tolerance to salt and alkali stress, *Leymus chinensis* populations on the southwestern Songnen Plain in northern China are threatened by increasing soil salinity and alkalinity. To explore the species' responses to saline-alkali stress, we grew it in substrates with varying concentrations of nitrogen (N) and phosphorus (P) while applying varying levels of saline-alkali stress (increasing in 14-, 17- or 23-day intervals). We measured the plants' contents of N and P, and the N:P ratio, and calculated their homeostasis indices ($H_N$, $H_P$ and $H_{N:P}$) under each nutrient and saline-alkali stress treatment. The N content was found to be more sensitive to saline-alkali stress than the P content. The N and P contents were highest and the N:P ratio was stable at pH 8.4. At both pH 8.1 and 8.4, $H_{N:P} > H_N > H_P$, but the indices and their relations differed at other pH values. Exposure to saline-alkali stress for the 14-day incremental interval had weaker effects on the plants. Rapid changes in salinity-alkalinity weakened both the positive effects of the weakly alkaline conditions (pH 7.5–8.4) and the negative effects of more strongly alkaline conditions (pH 8.7 or 9.3) on *L. chinensis*. When *L. chinensis* plants lack N, applying N fertilizer will be extremely efficient. The optimal concentrations of N and P appeared to be 16 and 1.2 mmol/L, respectively. When the *L. chinensis* plants were N- and P-limited, the specific growth rate correlated positively with N:P, when limited by N it correlated positively with the environmental N concentration, and when limited by P it was weakly positively correlated with the environmental P concentration.

# INTRODUCTION

Ecological stoichiometry (ES) is the study of the balances of elements and energy in ecosystems, which have profound effects on living organisms, their interactions, and associated ecological processes (*Cambardella & Elliott, 1993*; *Elser et al., 2000*; *Güsewell, 2004*; *Bradshaw, Kautsky & Kumblad, 2012*). ES theory focuses primarily on elements required by all living organisms, so it can be readily generalized across taxa and systems

Corresponding author
Yuefen Li, yfli@jlu.edu.cn

(*Sanders & Taylor, 2018*). A key concept is homeostasis: a system's capacity to maintain constant conditions internally when external conditions vary, a fundamental property of organisms (*Kooijman, 1995*; *Cooper, 2008*; *Halvorson et al., 2019*). Characterizing both the degrees and consequences of homeostasis is important for understanding responses to environmental changes (*Meunier, Malzahn & Boersma, 2014*; *Halvorson et al., 2019*). In ES-based analyses, the homeostasis concept is used to infer organisms' responses to changes in ratios of elemental resources, and predict their feedback effects on resources' availability through changes in the acquisition, storage, and release of nutrients, particularly limiting nutrients such as nitrogen (N) and phosphorus (P) (*Halvorson et al., 2019*). A general assumption is that stoichiometric homeostasis is stronger in heterotrophs than autotrophs (*Sterner & Elser, 2002*), so environmental stoichiometry is believed to be reflected more closely by plant and algal stoichiometry than by animals' stoichiometry. However, *Yu et al. (2011)* found that homeostatic regulation of N and P varies widely in vascular plants, so the patterns and processes involved are complex and far from fully understood.

Another core concept of ES is the growth rate hypothesis (GRH), which postulates that organisms' specific growth rates correlate positively with their phosphorus (P) contents because P is essential for the ribosomal RNA production needed to sustain growth (*Moody et al., 2017*) and for many other processes including photosynthesis, respiration, enzyme production, and generation of ATP (*Li et al., 2019*). Thus, organisms with high specific growth rates have high nutrient demands and (*inter alia*) low tissue C:nutrient ratios, low N:P ratios, and potentially competitive advantages in high-P environments, but disadvantages in low-P environments. Hence, plants' abilities to compete for nutrients depend on both their tissue nutrient contents and life history traits (*Mulder & Elser, 2009*; *González et al., 2010*; *Sanders & Taylor, 2018*).

ES has been widely applied in various areas of plant science, including in studies on the growth of individual plants, population dynamics, limiting elements, succession, and stability of vegetative communities (*Güsewell, 2004*). Soil is a key component of terrestrial ecosystems because it supports numerous ecological processes (*Normand et al., 2017*), plays crucial roles in plant growth, and directly affects plant communities' composition, stability, and succession (*Wardle, Walker & Bardgett, 2004*). N and P are key elements with profound effects on plants growth because they play major roles in microbial dynamics, litter decomposition, food webs, and the accumulation and cycling of nutrients in soil (*Elser et al., 2003*). Changes in plants' N:P ratios (*inter alia*) may reflect adaptations to environmental conditions (*Tang et al., 2018*), including (of particular interest here) salinity and alkalinity.

According to incomplete UNESCO and FAO statistics, the area of saline-alkali land is growing globally (*Kerr, 2017*) and amounts to $9.5438 \times 10^8$ $hm^2$, including $0.9913 \times 10^8$ $hm^2$ in China (*Li et al., 2017a*). This is a severe problem because salinization impairs seed germination, root extension, and plant development, resulting in land degradation, reductions in cultivated land quality, and limitation of agricultural development (*Wu et al., 2016*). Most saline-alkali soil in northeast China is on the Songnen Plain, and its area is increasing through grassland salinization (*Li, 2000*). The dominant species in saline-alkali grassland on the Songnen Plain is *Leymus chinensis (L. chinensis)*, which can provide high

yields of high quality gramineous forage if the salinity and alkalinity are not too severe. It can play an important role in restoring saline-alkali land (*Liu et al., 2014*), but there is a need to reduce the salinity and alkalinity of affected grassland and improve the species' yield and stress tolerance. It is therefore important to study the ecological stoichiometric homeostasis of N and P, associated ecological processes, and the effects of saline-alkali stress on the growth rate and survival strategy of *L. chinensis*.

Previous research on saline-alkali tolerance in *L. chinensis* has mainly focused on physiological processes (such as the osmoregulatory roles of ions, organic acids, soluble sugars and other substances) and morphological traits (such as tiller buds, root distribution, and rhizomes' internode length and branching angles) (*Zheng et al., 2017*). C, N, and P dynamics in soil-*L. chinensis* systems have also been addressed (*Li et al., 2018*). The secret of *L. chinensis* adapting to saline-alkali stress has been discussed deeply from many perspectives. However, soil salinization is a long-term dynamic process, and there is relatively little information on relations between the effects of saline-alkaline stress (and of variation in its duration) and ecological stoichiometric characteristics. Also, few studies point out a certain range in which *L. chinensis* can adapt to lasting saline-alkali stress. We make the hypothesis that the adaptability of *L. chinensis* under different saline-alkali stress conditions is different (positive effects or negative effects) and there is a certain range. Here, we demonstrate that the ecological stoichiometric characteristics of *L. chinensis* are influenced by the duration of saline-alkali stress and the available concentrations of both N and P.

## MATERIALS AND METHODS

### The experimental site

We performed experiments in an outdoor terrace on the campus of the College of Earth Sciences, Jilin University, Changchun, Jilin Province, northeast China. This is in the north temperate continental climatic zone, which has four distinct seasons. The mean annual temperature and precipitation are 4.8 °C and 567 mm, respectively, and mean winter and summer temperatures are -14 and 24 °C, respectively (*Zhang et al., 2017*).

### Experimental design
#### Experimental preparation
Experiments were performed from August 7 to November 6, 2017. To minimize interference from impurities in the soil, the substrate used was fine nutrient-poor sand, sieved with 10 and 75 mesh sieves and thoroughly washed with distilled water to minimize its nutrient content. To avoid effects of different soil loadings on growth of experimental plants, the sand was air-dried and equal portions were placed in plastic flowerpots (diameter 30 cm, height 23 cm). The portions were roughly equivalent to two-thirds of the pots' volume. *L. chinensis* seeds were immersed in water for 48 h and air-dried in the dark to avoid dormancy and increase the germination rate, then about 100 seeds were sown evenly in each flowerpot. About seventy percent of the seeds survived, and all pots were similar with respect to the number, size, and vitality of surviving seedlings.

### N and P addition

After the seeds germinated, a 200 ml portion of Hoagland's nutrient solution with adjusted N and P concentrations was added to each pot once every two days. The elemental composition of Hoagland's solution is described in detail elsewhere (*Li et al., 2018*). In five sets of nine pots, the N concentration was adjusted using ammonium nitrate to establish five N concentration treatments (2, 4, 8, 16, and 24 mmol/L) in which the concentration of P was fixed at one mmol/L. In another five sets of nine pots, the P concentration was adjusted using potassium dihydrogen phosphate to establish five P concentration treatments (0.3, 0.6, 1.2, 2.4, and 4.8 mmol/L) in which the concentration of N was fixed at 14 mmol/L. The experiment thus involved 10 nutrient treatments and a total of 90 pots (10 treatments ×3 replicates ×3 salinity stress intervals) in a randomized complete block design. The N and P concentration gradients were set on the basis of previously reported results (*Yu et al., 2011*).

### Saline-alkali stress

On 7 Aug 2017, after the *L. chinensis* plants had grown to an average height of about 15 cm, plants in triplicate pots were subjected to the saline-alkali stress treatments shown in Table 1, by adding $NaHCO_3$ to the previously described treatment solution to simulate various degrees of land salinization encountered in the western regions of Jilin Province. To avoid excessively stressing the plants, the pH was increased at intervals of 14, 17, or 23 days. The pH was raised from 7.5 to 8.1, and then to 8.4, 8.7, and finally 9.3. Aboveground parts of *L. chinensis* plants subjected to these treatments were harvested (cutting from the ground level) according to the schedule shown in the Table 1.

### Sample collection and testing methods

Aboveground parts of *L. chinensis* plants were cut according to the harvesting schedule shown in Table 1. The samples were rinsed with distilled water, dried with absorbent paper, deactivated for 15 min in a dry air oven (105 °C) and dried for 12 h at 65 °C to eliminate water completely. After cooling the samples to room temperature, they were weighed, crushed, screened with a 100 mesh sieve, and finally quartered. The resulting sub-samples were then packed in sealed bags for analysis.

Total N and total P in the *L. chinensis* samples were respectively determined by Chinese standard methods LY/T 1269-1999 (using a SAN++ Continuous Flow Analyzer; Skalar, Netherlands) and LY/T 1270-1999 (using an ICPS-7500 inductively coupled plasma atomic emission spectrometer; Shimadzu, Kyoto, Japan).

### Data analysis

The acquired data were statistically analyzed using SPSS 24 (SPSS Inc., USA). Homeostasis indices were calculated using the stoichiometric homeostasis model $y = cx^{\frac{1}{H}}$. Here, $x$ is the measured content of an element in the soil, $y$ is the measured content of the same element in *L. chinensis*, $c$ is a constant coefficient, and $H$ is the homeostasis index. The results were visualized using Sigmaplot 12.5 (Systat Software, Inc.). We calculated specific growth rates based on recorded changes in dry mass and instantaneous growth rates using the equation $u = \ln(M_t/M_0)/t$, where $u$ is the specific growth rate (day$^{-1}$), $M_t$ and $M_0$, are the final

**Table 1** Saline-alkali stress treatments, including durations of exposure to each pH, dates of exposure, and harvest dates.

| N-P level | Duration (days) | pH | Dates of exposure | Harvest date |
|---|---|---|---|---|
| | | 7.5 | $2017/8/07 - 2017/8/17$ | 2017/8/18 |
| | | 8.1 | $2017/8/18 - 2017/8/31$ | 2017/9/01 |
| | Incremental interval 14 days | 8.4 | $2017/9/01 - 2017/9/14$ | 2017/9/15 |
| | | 8.7 | $2017/9/15 - 2017/9/28$ | 2017/9/29 |
| $3 \times$N1-P | | 9.3 | $2017/9/29 - 2017/10/12$ | 2017/10/13 |
| $3 \times$N2-P | | 7.5 | $2017/8/07 - 2017/8/22$ | 2017/8/23 |
| $3 \times$N3-P | | 8.1 | $2017/8/23 - 2017/9/08$ | 2017/9/09 |
| $3 \times$N4-P | Incremental interval 17 days | 8.4 | $2017/9/09 - 2017/9/25$ | 2017/9/26 |
| $3 \times$N5-P | | 8.7 | $2017/9/26 - 2017/10/12$ | 2017/10/13 |
| $3 \times$N-P1 | | 9.3 | $2017/10/13 - 2017/11/05$ | 2017/11/06 |
| $3 \times$N-P2 | | 7.5 | $2017/8/07 - 2017/8/27$ | 2017/8/28 |
| $3 \times$N-P3 | | 8.1 | $2017/8/28 - 2017/9/19$ | 2017/9/20 |
| $3 \times$N-P4 | Incremental interval 23 days | 8.4 | $2017/9/20 - 2017/10/12$ | 2017/10/13 |
| $3 \times$N-P5 | | 8.7 | $2017/10/13 - 2017/11/05$ | 2017/11/06 |
| | | 9.3 | / | No Harvest |

**Notes.**

There were 10 nutrient treatments and a total of 90 pots were used in the experiment. The salt-alkali stress experiment at incremental interval 23 days with the pH of 9.3 was not carried out due to the sudden drop of local temperature at the end of October.

and initial dry mass, respectively, and $t$ is experiment duration in days. Details of several statistical analyses are included in Appendix 1 and Appendix 2.

# RESULTS

## Effects of incremental increases in saline-alkali stress on N, P contents and N:P ratios

The 14-day stress increment yielded the highest N and P contents in the plants when the N concentration in the substrate was low (2–8 mmol/L), but not when the N concentration in the substrate was higher (16 or 24 mmol/L). The variation in the N and P contents, and the N:P ratio, was similar under all three stress intervals: the N content and N:P ratio were highest with 17-day intervals, while the P content was highest with 14-day intervals (Fig. 1).

## Homeostasis characteristics of *L. chinensis*

The homeostasis index of N ($H_N$) in *L. chinensis* ranged from 2.35 to 7.25, and first rose then fell as salinity-alkalinity increased, independently of interval length (Fig. 2). Under the 14-, 17- and 23-day intervals, $H_N$ ranges were 3.10–7.25, 2.35–5.36 and 3.19–3.87, respectively. It was consistently highest at pH 8.4, and higher at pH 8.1 and 8.7 than at pH 7.5 and 9.3 (Table 2). Under the 14-day stress intervals, the changes in $H_N$ were moderate, and $H_N$ was high at all pH values. The index was lower at both pH 8.1 and 8.7 under the 17- and 23-day intervals.

Under our treatments, $H_P$ of *L. chinensis* ranged from 2.60 to 5.33. Under all three saline-alkali stress intervals, it first rose and then declined (Fig. 2). Under the 14-, 17-,

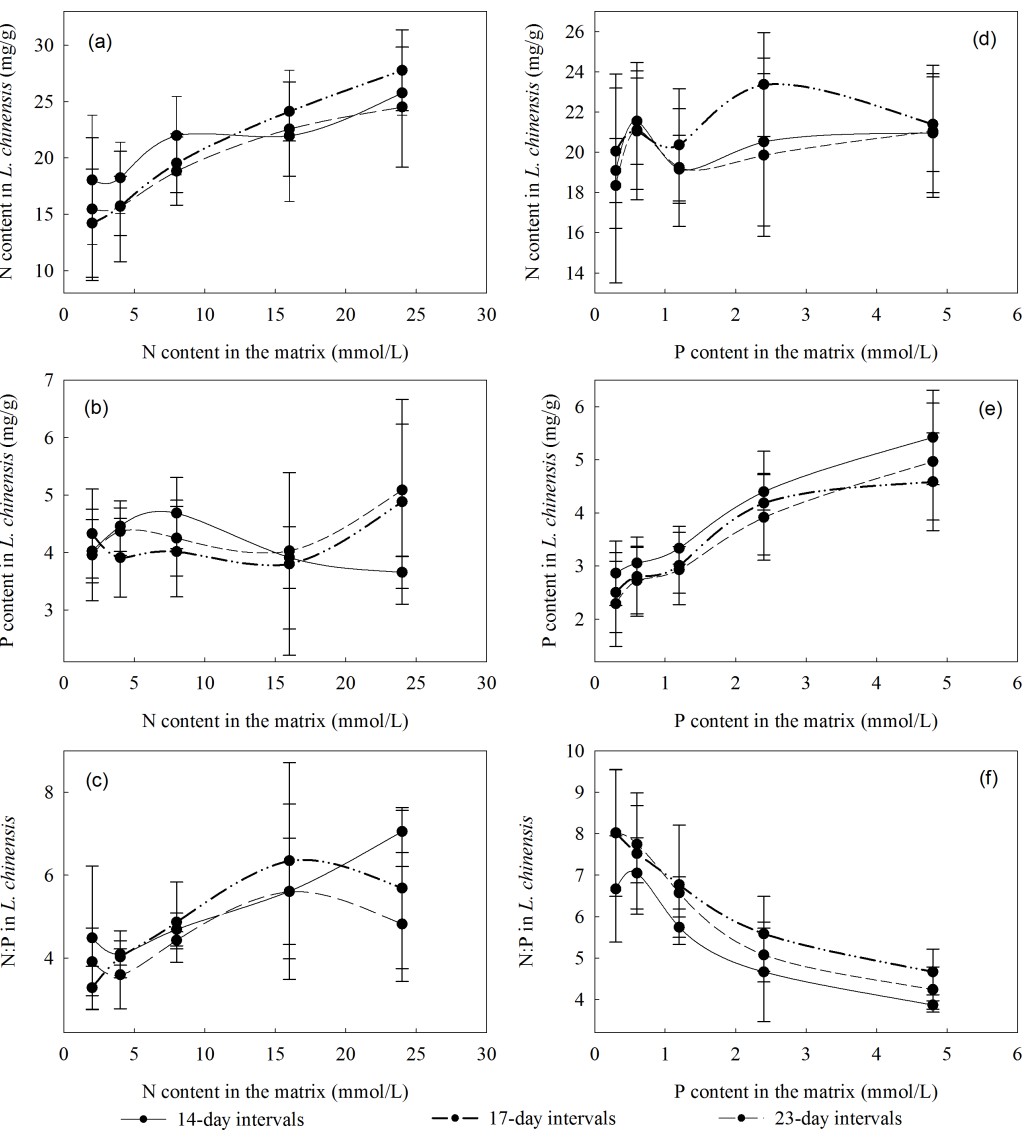

**Figure 1 Relationships between the N and P contents, and N:P ratios of *L. chinensis* plants and the intervals between saline-alkali stress increments.** (A) N content in *L. chinensis* plotted against the N content (2, 4, 8, 16 and 24 mmol/L) applied to matrix in the N addition experiment, (B) P content in *L. chinensis* plotted against the N content (2, 4, 8, 16 and 24 mmol/L) applied to matrix in the N addition experiment, (C) N:P in *L. chinensis* plotted against the N content (2, 4, 8, 16 and 24 mmol/L) applied to matrix in the N addition experiment, (D) N content in *L. chinensis* plotted against the P content (0.3, 0.6, 1.2, 2.4 and 4.8 mmol/L) applied to matrix in the P addition experiment, (E) P content in *L. chinensis* plotted against the P content (0.3, 0.6, 1.2, 2.4 and 4.8 mmol/L) applied to matrix in the P addition experiment, (F) N:P in *L. chinensis* plotted against the P content (0.3, 0.6, 1.2, 2.4 and 4.8 mmol/L) applied to matrix in the P addition experiment. Solid lines, dash-dot-dot lines and medium dash lines refer to intervals of 14, 17, and 23 days, respectively. Error bars are standard deviations.

and 23-day saline-alkali stress intervals, the ranges of $H_P$ were 2.84–5.33, 2.65–4.90 and 2.60–4.10, respectively, and the highest values of $H_P$ occurred at pH 8.1, 8.7, and 8.1, respectively (Table 2). At pH 8.1, $H_P$ was significantly higher under 14-day intervals than

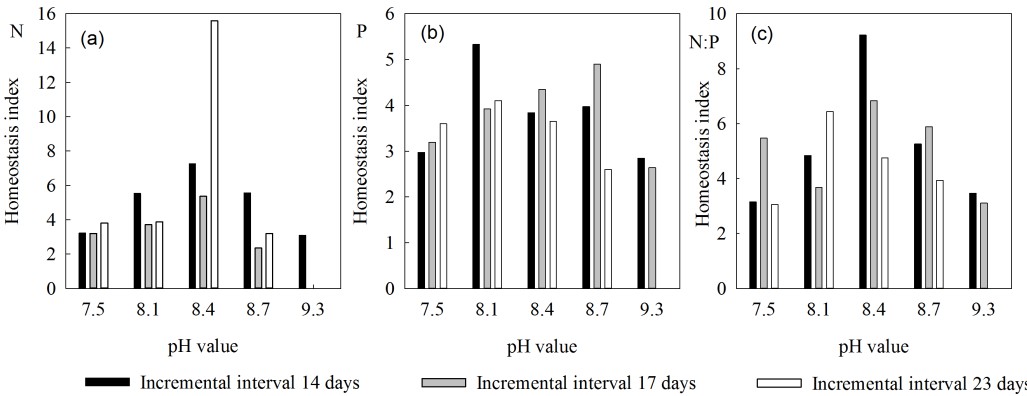

**Figure 2  Homeostasis indices of *L. chinensis* at the indicated pH values and saline-alkali stress incremental intervals.** (A) The homeostasis indices of N (HN) in *L. chinensis*, (B) The homeostasis indices of N (HP) in *L. chinensis*, (C) The homeostasis indices of N (HN:P) in *L. chinensis*. Black columns, gray columns and white columns refer to intervals of 14, 17, and 23 days, respectively.

**Table 2  Homeostasis indices ($H$) of aboveground parts of *L. chinensis*, and their linear regression correlation coefficients ($R^2$), under the indicated saline-alkali stress treatments.**

| pH | Incremental interval 14 days | | Incremental interval 17 days | | Incremental interval 23 days | | |
|---|---|---|---|---|---|---|---|
| | $H_N$ | $R^2$ | $H_N$ | $R^2$ | $H_N$ | $R^2$ | Average $H_N$ |
| 7.5 | 3.23 | 0.93 | 3.20 | 0.92 | 3.81 | 0.93 | 3.41 |
| 8.1 | 5.53* | 0.22* | 3.70 | 0.98 | 3.87 | 0.95 | 4.37 |
| 8.4 | 7.25 | 0.95 | 5.36 | 0.83 | 15.58* | 0.34* | 6.31 |
| 8.7 | 5.57 | 0.64 | 2.35 | 0.96 | 3.19 | 0.93 | 3.70 |
| 9.3 | 3.10 | 0.89 | — | — | — | — | 3.10 |
| | $H_P$ | $R^2$ | $H_P$ | $R^2$ | $H_P$ | $R^2$ | Average $H_P$ |
| 7.5 | 2.97 | 0.99 | 3.19 | 0.99 | 3.60 | 0.92 | 3.25 |
| 8.1 | 5.33 | 0.80 | 3.92 | 0.96 | 4.10 | 0.79 | 4.45 |
| 8.4 | 3.84 | 0.70 | 4.35 | 0.84 | 3.65 | 0.85 | 3.95 |
| 8.7 | 3.97 | 0.96 | 4.90 | 0.95 | 2.60 | 0.94 | 3.82 |
| 9.3 | 2.84 | 0.99 | 2.64 | 0.90 | — | — | 2.74 |
| | $H_{N:P}$ | $R^2$ | $H_{N:P}$ | $R^2$ | $H_{N:P}$ | $R^2$ | Average $H_{N:P}$ |
| 7.5 | 3.15 | 0.96 | 5.47 | 0.82 | 3.06 | 0.95 | 3.89 |
| 8.1 | 4.84 | 0.82 | 3.68 | 0.93 | 6.44 | 0.79 | 4.98 |
| 8.4 | 9.22* | 0.31* | 6.83 | 0.86 | 4.76 | 0.89 | 6.94 |
| 8.7 | 5.27 | 0.84 | 5.89 | 0.91 | 3.94 | 0.84 | 5.03 |
| 9.3 | 3.46 | 0.86 | 3.11 | 0.96 | — | — | 3.29 |

**Notes.**
A dash (—) indicates that no valid data were obtained, and "*" indicates an outlier.

under longer intervals. At pH 8.4, it was highest under 17-day intervals, and at pH 8.7 it declined in the order 17-day >14-day >23-day intervals.

Plants have strong self-regulating mechanisms, and generally keep their N:P ratios within a narrow range by adjusting their N and/or P contents in response to environmental

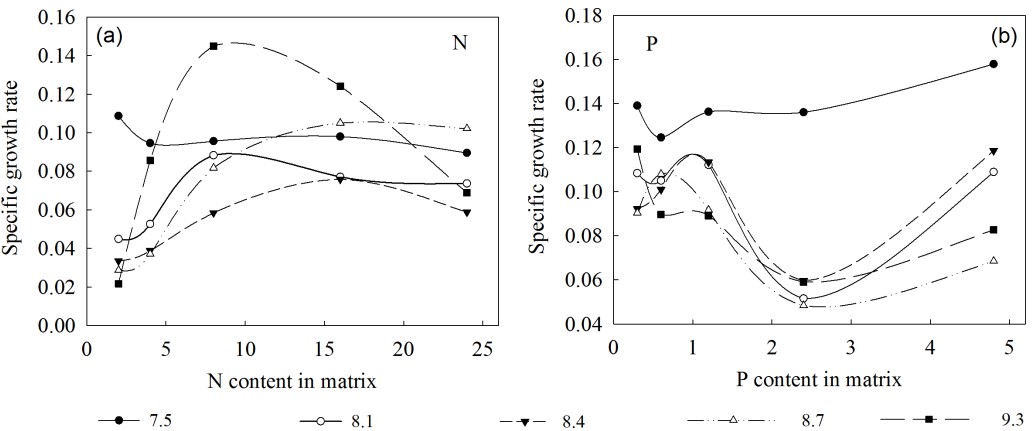

**Figure 3  Specific growth rates of *L. chinensis* at the indicated environmental N and P concentrations, and pH.** (A) The specific growth rates of *L. chinensis* plotted against the N content (2, 4, 8, 16 and 24 mmol/L) applied to matrix in the N addition experiment, (B) The specific growth rates of *L. chinensis* plotted against the P content (0.3, 0.6, 1.2, 2.4 and 4.8 mmol/L) applied to matrix in the P addition experiment. Solid circles, hollow circles, solid triangles, hollow triangles and solid rectangles refer to pH of 7.5, 8.1, 8.4, 8.7, and 9.3, respectively.

changes. For example, the $H_{N:P}$ of *L. chinensis* varied from 3.06 to 9.23 in plants exposed to different P concentrations, while its ranges under the 14-, 17- and 23-day stress interval regimes were 3.15–9.23, 3.11–6.83 and 3.06–6.44, respectively, and its peak values occurred at pH 8.4, 8.4, and 8.1, respectively (Table 2). *L. chinensis* plants are thus able to effectively regulate their *N:P* ratio in the pH range 8.1 to 8.7, but less so at pH 7.5 and 9.3.

## The specific growth rate of *L. chinensis*

The specific growth rate of the plants rapidly increased then decreased as the N concentration in the substrate increased, peaking at 16 mmol/L (Fig. 3). This indicates that when *L. chinensis* plants lack N, initial N addition will be extremely efficient, but saturating or excessive amounts will not promote (and may even hinder) further growth. Conversely, the plants' specific growth rate first increased, then decreased, and then increased again with increases in environmental P concentration. Specifically, it was lowest at a P concentration of 2.4 mmol/L, and higher at both 1.2 and 4.8 mmol/L. Thus, high addition of P is beneficial for plant growth but may lead to wastage of resources. Therefore, the optimum P concentration under our experimental conditions was 1.2 mmol/L.

In summary, the threshold concentrations of N and P in the substrate solution at which the biomass of *L. chinensis* began to decrease (or stopped increasing) were 16 and 1.2 mmol/L, respectively. Moreover, plants exposed to 1.2–4.8 mmol/L P were mainly restricted by N, while those exposed to 16 and 24 mmol/L N were mainly restricted by P, and those exposed to lower concentrations of P and N were restricted by P and N, respectively (Fig. 4). According to the results of the correlation analysis, the specific growth rate of *L. chinensis* and the plant N:P ratio is positively correlated in both N-limited and P-limited environments, and the correlation is slightly stronger under P-restricted conditions, but neither of them passed the significance test. In the N and P co-limited environment, the

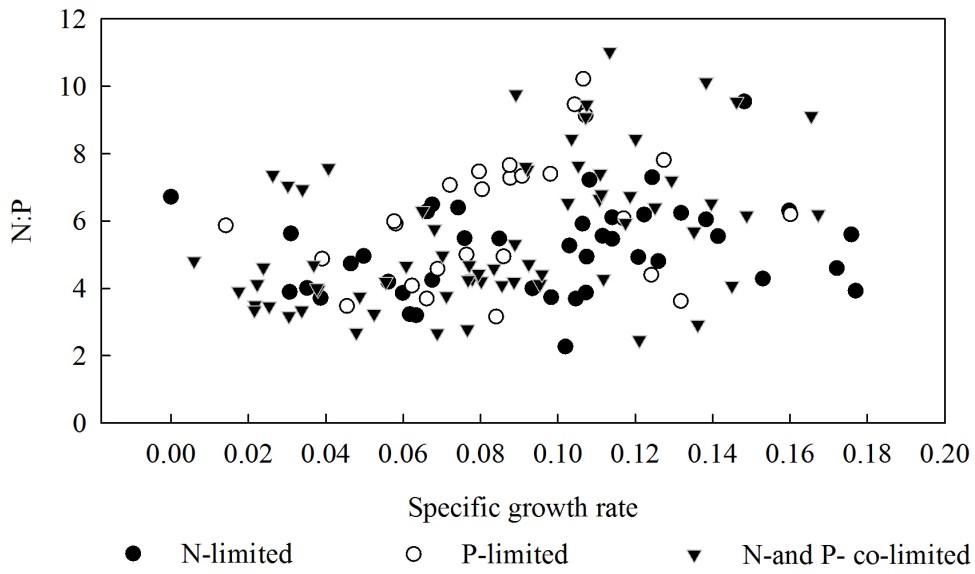

**Figure 4** **Relationship between the specific growth rate and the N:P ratio in the aboveground parts of** *L. chinensis.*

specific growth rate of *L. chinensis* has a positive correlation with the plant N:P ratio, and the correlation coefficient is 0.46, which is significant at the level of 0.01.

## DISCUSSION

### N and P contents and the N:P ratio in *L. chinensis*

N and P are key nutrients that play major roles in myriad processes in plants and have important effects on one-another's uptake and metabolism (*Han et al., 2009*). N:P ratios in plants fluctuate within relatively narrow ranges, and variations in their N and P contents are restricted by homeostatic mechanisms. Although P and N contents can influence each other, N affects P contents more strongly than vice versa, putatively due to the overall higher abundance of N in plants (*Yu, 2005*). We found that *L. chinensis* plants absorbed limited amounts of P from soils with a low N content, and changes in the soil's N contents between 2 and eight mmol/L had minor effects on the P content of their aboveground parts. However, their P contents were substantially increased by higher N concentrations in the soil. Similarly, we found that N contents of aboveground parts of *L. chinensis* are influenced by the environmental P concentration: they changed greatly when the soil's P content was low but remained very similar at high P concentrations. In summary, P content is the main constraint on plant growth when the soil's P content is low, but N content becomes the main constraint of plant growth when the P content is high.

Our experiment examined saline-alkaline environments with different pH values. The N and P contents in aboveground parts of *L. chinensis* were both high at pH 8.4, indicating that the plants maintained strong control over their N:P ratios at this pH and confirming the previously reported finding that this pH promotes growth of *L. chinensis* (*Li et al., 2018*). *L. chinensis* can resist mild pH stress but it is inhibited by strong saline-alkali stress

(*Yan et al., 2006*). At high pH (8.7 and 9.3), we found that its P content was relatively high when there were high soil concentrations of either N (16 or 24 mmol/L) or P (2.4 or 4.8 mmol/L). However, plants need more P to cope with pH stress. Plants' nutrient adsorption is related to pH, together with indications that luxury uptake of N and P under moderate pH stress enables some plants (e.g., *Rhizoma acori graminei* and *Lythrum salicaria*) to resist strong pH stress (*Cheng et al., 2017*). The N and P contents in aboveground parts of *L. chinensis* we observed at various pH values indicate that N contents in plants are more sensitive to saline conditions than their P contents.

## Homeostasis characteristics of N, P, and N:P in *L. chinensis*

Homeostasis theory postulates that plants can adjust their growth rate, and rates of both resource allocation and utilization in mechanisms that maintain their internal homeostasis (*Mendez & Karlsson, 2005*). Homeostasis is stronger in animals than in plants, and in higher plants than lower plants (*Zeng et al., 2016*), suggesting that homeostatic mechanisms have become more powerful over the course of evolutionary history. We found that the homeostasis indices for N content, P content and the N:P ratio in *L. chinensis* were within reported ranges for algae, fungi (*Sterner & Elser, 2002*), and animals (*Yu, 2005*; *Karimi & Folt, 2006*). On the basis of experiments on Inner Mongolian grassland and sand culture, *Yu (2009)* concluded that aboveground parts of *L. chinensis* have slightly higher $H_N$, $H_P$ and $H_{N:P}$ values (5.88–8.80, 3.37–6.67 and 4.49–9.46, respectively) than those observed in our study, possibly due to differences in climatic conditions.

We also found that all three homeostasis indexes first increased then decreased with increases in pH. At pH 8.1 and 8.4, they were all high, showing that *L. chinensis* can maintain stable levels of nutrients in weakly alkaline environments. However, the indices were lower under more strongly alkaline conditions (pH > 8.7), indicating that such conditions have toxic effects. This is consistent with previous observations that saline and alkaline stress has both physiological and biochemical effects on roots and leaves of *L. chinensis* (*Liu et al., 2014*).

Another finding is that $H_{N:P}>H_N>H_P$, indicating that the stronger homeostasis of N content than of P content in plants is mainly due to their significantly higher N content. Accordingly, homeostasis in zooplankton is reportedly highest for macronutrients, followed by micronutrients, and lowest for non-essential elements (*Karimi & Folt, 2006*). Since high N contents in plant tissues promote high P contents and plants' N:P ratios vary less than their N and P contents, the N:P ratio may be subject to stronger homeostasis than either N or P individually (*Sterner & Elser, 2002*). Plants maintain N:P stability by their adjusting resource allocation and utilization of resources. The degree of stability may reflect their environmental adaptability, so the N:P ratio appears to be more important than N and P contents in this respect.

The plants' homeostasis is related to the duration of the incremental increases in salinity-alkalinity stress. With 14-day intervals (rapid changes in alkalinity), the positive effects of a weakly alkaline environment (pH 7.5–8.4) on leaves of *L. chinensis* are stronger than those observed with longer intervals: the homeostasis indices of the aboveground parts are relatively high, and further increases in alkalinity (to pH 8.7 or 9.3) induce relatively

little damage to leaves. H $_{N:P}$ and H $_P$ are strongly affected by changes in the duration of the incremental increases. For example, the variation in H $_{N:P}$ of aboveground parts is higher with 23-day intervals than with either 14- or 17-day intervals (rising to a peak and then quickly falling with increases in pH, as shown in Fig. 2).

## Specific growth rate

Specific growth rates are crucial indicators of organisms' adaptation to (and thus ability to survive and reproduce within) an environment. Clearly, variations in growth capacity can be linked to variations in elemental demand (*Moody et al., 2017*). Our results confirm previous findings (and intuitive expectations) that adding N can significantly enhance growth of N-deficient plants. However, they also show that adding N can inhibit plants' growth if the environmental content of N exceeds the plants' requirements. The optimal N and P contents in the soil under our experimental conditions were 16 and 1.2 mmol/L, respectively. Adding excess P will not have inhibitory effects, and may even further stimulate growth slightly, but would be a waste of resources.

Plants can absorb elements in soils selectively. Moreover, The N:P ratio has a relatively complex relationship with specific growth rates: they are positively correlated at low specific growth rates, but negatively correlated once the specific growth rate exceeds a certain threshold (*Agren, 2004*). Additionally, specific growth rates of belowground parts of plants were positively correlated with N:P under N constraints, but negatively correlated with N:P under P constraints (*Yu, 2009*). In our experiment, growth of *L. chinensis* was restricted by both N and P. The specific growth rate of aboveground parts was positively related to N:P under N constraint or P constraint, but the positive correlation was strengthened under the co-constraint of N and P. This is consistent with the conclusions of Yu et al. with respect to the effects of N constraint, but not those of P constraint. We speculate that this was due to the plant's generally low specific growth rate in the experiment, and also possibly to effects of salinity-alkalinity on the relationships between plant growth and other environmental variables. Although effects of plants' nutrient storage on their specific growth rates have been discussed (*Sterner & Elser, 2002*; *Yu, 2009*), the relationship between their specific growth rates and C:N:P ratios clearly requires further study.

According to the specific growth rate hypothesis, high amounts of ribosomal RNA (rRNA), and thus P, are needed to synthesize the large quantities of proteins required to sustain high specific growth rates (*Sterner & Elser, 2002*). Therefore, organisms with high specific growth rates have relatively high P contents and low N:P ratios. This hypothesis is supported by both theoretical considerations and empirical observations of zooplankton, arthropods, and bacteria (*Elser et al., 2003*; *Watts et al., 2006*; *Hessen et al., 2007*). However, the relationships may be more complex in higher plants. Accordingly, we observed the positive correlation between the specific growth rate and N:P ratio of *L. chinensis*, rather than the negative correlation predicted by the hypothesis. This may have been because we monitored adult *L. chinensis* plants rather than juveniles, and/or because the specific growth rate of the studied plants was strongly influenced by the variation of several environmental factors (N content, P content, and pH) and thus does not reflect their intrinsic potential specific growth rates.

## CONCLUSION

The findings presented here demonstrate that *L. chinensis* has the homeostasis ability under a certain degree of salinity-alkalinity stress. And the N content of *L. chinensis* is more sensitive to the environmental pH than its P content. At substrate pH values of 8.4 and 8.7, *L. chinensis* possesses good environmental adaptability. In particular, at a substrate pH of 8.4, *L. chinensis* were well able to control their contents of N and P as well as the N:P ratio. Weak alkalinity (pH 7.5–8.4) is beneficial for growth and N accumulation in *L. chinensis*, but more strongly alkaline conditions (pH 8.7 or 9.3) inhibit its growth. At pH values above 8.7, the interval between stress increments clearly affected the plants' contents of N and P as well as the N:P ratio. The relationship between the specific growth rate and N:P ratio may become more complex (not simple linear) because of the salinity-alkalinity stress.

## ACKNOWLEDGEMENTS

We thank Chenhao Cao, lunian Gao and Xiaowei Han for their assistance during plant cultivation and harvesting. We are grateful to the Associate Editor and one anonymous referee for providing valuable comments.

### Funding

This work was supported by the Science and Technology Strategy and Planning Research of Jilin Science and Technology Department (grant no. 20200101119FG). The funders had no role in study design, data collection and analysis, decision to publish, or preparation of the manuscript.

### Grant Disclosures

The following grant information was disclosed by the authors:
Science and Technology Strategy and Planning.
Jilin Science Technology Department:  20200101119FG.

### Competing Interests

The authors declare there are no competing interests.

### Author Contributions

- Shujie Li conceived and designed the experiments, analyzed the data, authored or reviewed drafts of the paper, and approved the final draft.
- Yujin Huang performed the experiments, analyzed the data, prepared figures and/or tables, and approved the final draft.
- Yuefen Li conceived and designed the experiments, performed the experiments, authored or reviewed drafts of the paper, and approved the final draft.

## Data Availability

The raw measurements are available in the Supplemental Files.

## Supplemental Information

Supplemental information for this article can be found online at http://dx.doi.org/10.7717/peerj.10768#supplemental-information.

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
