# Peer review of "Homeostatic responses and growth of Leymus chinensis under incrementally increasing saline-alkali stress"

_PeerJ, doi:10.7717/peerj.10768_

## Round 0.1 · original submission · Major Revisions

Authors should take into account the comments of the reviewers and prepare the manuscript in accordance with them.

Reviewer 1 ·

Basic reporting

no comment

Experimental design

no comment

Validity of the findings

no comment

Additional comments

1.-it was unclear to me in your methods how exactly you harvest. Did you clip to some residual height, or did you clip to ground level? Since residual height related to L. chinensis regrowth.

2.- Did harvest frequency disturb the effect of pH on L. chinensis growth? If yes, then it seems like you can't make any conclusions about the effect of the saline-alkali stress, because when the pH reached to 9.3 in 14-day salinity stress intervals, L. chinensis had been harvested 4 times (Table 1).

3.- Line 99: what did you mean adulthood? If L. chinensis had grown to adulthood on 7 August, did the leaf wilting when you harvested it on 28 August in 23-day salinity stress intervals, then how did you handle them?

·

Basic reporting

The study provides information pertaining to homeostatic responses and growth of Leymus chinensis under incrementally increasing saline-alkali stress. In abstract a meaningful conclusion may be added to make it more informative but brief. Literature cited is sufficient; however, more literature may be added to strengthen the relationship of saline-alkali with nutrients (NP) on the basis of ecological stoichiometric and growth model in introduction. The results are relevant and prove the hypothesis.

Experimental design

The experimental design has not been made clear and needs more details. However, statistical data analysis is well.

Validity of the findings

The findings are novel and add to existing knowledge.

Reviewer 3 ·

Basic reporting

The research analyzed the homeostatic responses and growth to incrementally increasing saline-alkali stress in Leymus chinensis. It is providing some new supports for current theoretical research. However, the paper exists a lot of flaws, thus need to done a major revision.

Abstract: the present is not good, need to deep revise it. Detail comments are seen in attached pdf file.

Introduction: introduction needs more detail. I suggest that you add the description at lines 59- 61 to provide more justification to expand upon the knowledge gap being filled.
In addition, the research no clear hypothesis. Need to add it.

Experimental design

Methods described with un-sufficient detail and information to replicate. Detail as follows: 1. Where the plants were grow during experiment? Growing house? Need details; 2. Line 78-79. Why you sealed bottom hole? It is will lead the soil very salty; 3. Most importantly, line 88-89, Hoagland’s nutrient solution contains KNO3, NH4H2PO4, and the N concentration is about 4 times than field condition. It is will decrease your N treatment effect in experiment. Need to explain the relationship between of them; Same question for P.

Validity of the findings

Data analysis: 1. which model you used in analysis? Need details; 2. Add error bars in all figures.

Results: only presented N, P, N:P, why no any information about biomass or growth rate et al?

Discussion: Should revise it based on revised results.

Conclusion: not good. Need to re-write it.

Refs: too much. 30 enough.

Additional comments

The research analyzed the homeostatic responses and growth to incrementally increasing saline-alkali stress in Leymus chinensis. It is providing some new supports for current theoretical research. However, the paper exists a lot of flaws, thus need to done a major revision.

Abstract: the present is not good, need to deep revise it. Detail comments are seen in attached pdf file.

Introduction: introduction needs more detail. I suggest that you add the description at lines 59- 61 to provide more justification to expand upon the knowledge gap being filled.
In addition, the research no clear hypothesis. Need to add it.

Methods: 1. Where the plants were grow during experiment? Growing house? Need details; 2. Line 78-79. Why you sealed bottom hole? It is will lead the soil very salty; 3. Most importantly, line 88-89, Hoagland’s nutrient solution contains KNO3, NH4H2PO4, and the N concentration is about 4 times than field condition. It is will decrease your N treatment effect in experiment. Need to explain the relationship between of them; Same question for P.

Data analysis: 1. which model you used in analysis? Need details; 2. Add error bars in all figures.

Results: only presented N, P, N:P, why no any information about biomass or growth rate et al?

Discussion: Should revise it based on revised results.

Conclusion: not good. Need to re-write it.

Refs: too much. 30 enough.

Annotated reviews are not available for download in order to protect the identity of reviewers who chose to remain anonymous.

---

## Round 0.2 · Minor Revisions

Authors need to reasonably respond to the reviewer's comments and take these comments into account in the manuscript.

Reviewer 3 ·

Basic reporting

Most of comments have been addressed in the revision. Language also fine. However, next few place need to revised. 1. Abstract: the first sentence not good. Leymus has grown salinty and alkalinity environment thousands and thousands years, that is, it has well adopted sal-alk environment. Thus, we cannot concluded that Leymus threatened by increasing.... 2. No any hypotheses were found in introduction. I strong suggest authors add it in revision.

Experimental design

1. Experimental site description is useless in the study. The author should give some details information where the plant was placed. If placed in greenhouse, what about temperature, moisture, and PAR?
2. Lack of experimental design method like RCBD and so on. Need to add it in next revision.

Validity of the findings

no comment

Additional comments

Most of comments have been addressed in the revision. Language also fine. However, next few place need to revised. 1. Abstract: the first sentence not good. Leymus has grown salinty and alkalinity environment thousands and thousands years, that is, it has well adopted sal-alk environment. Thus, we cannot concluded that Leymus threatened by increasing.... 2. No any hypotheses were found in introduction. I strong suggest authors add it in revision. 3. Experimental site description is useless in the study. The author should give some details information where the plant was placed. If placed in greenhouse, what about temperature, moisture, and PAR? 4. Lack of experimental design method like RCBD and so on. Need to add it in next revision.

---

## Round 0.3 · accepted · Accept

The authors have made all the necessary improvements in the work, and the manuscript can be published.